# Diversity Sampling Regularization for Multi-Domain Generalization

**Lakpa Tamang**                                    *l.tamang@deakin.edu.au*
*Deakin University*

**Mohamed Reda Bouadjenek**                          *reda.bouadjenek@deakin.edu.au*
*Deakin University*

**Richard Dazeley**                                 *richard.dazeley@deakin.edu.au*
*Deakin University*

**Sunil Aryal**                                     *sunil.aryal@deakin.edu.au*
*Deakin University*

**Reviewed on OpenReview:** *https://openreview.net/forum?id=nXqMt7X2RX*

## Abstract

Domain Generalization (DG) seeks to create models that can successfully generalize to new, unseen target domains without the need for target domain data during training. Traditional approaches often rely on data augmentation or feature mixing techniques, such as MixUp; however, these methods may fall short in capturing the essential diversity within the feature space, resulting in limited robustness against domain shifts. In this research, we revisit the importance of diversity in DG tasks and propose a simple yet effective method to improve DG performance through diversity-sampling regularization. Specifically, we calculate entropy values for input data to assess their prediction uncertainty, and use these values to guide sampling through Determinantal Point Process (DPP), which prioritizes selecting data subsets with high diversity. By incorporating DPP-based diversity sampling as a regularization strategy, our framework enhances the standard Empirical Risk Minimization (ERM) objective, promoting the learning of domain-agnostic features without relying on explicit data augmentation. We empirically validate the effectiveness of our method on standard DG benchmarks, including PACS, VLCS, OfficeHome, TerraIncognita, and DomainNet, and through extensive experiments show that it consistently improves generalization to unseen domains and outperforms widely used baselines and S.O.T.A without relying on any task-specific heuristics. Our implementation is available at: `https://github.com/lakpa-tamang9/domaingen`

## 1 Introduction

Machine learning models have demonstrated tremendous progress across diverse applications, yet they continue to struggle when confronted with changes in data distribution. Due to the inherent nature of training such models with *independent and identical distribution* (i.i.d) hypothesis, their generalization performance on unseen test data in a heterogeneous distribution space is significantly impacted (Tamang, 2024). This phenomenon in machine learning is termed as the Out-of-Distribution (OOD) problem, where the change in features/covariates of the data occurs as a result of domain shift or environmental change. Figure 1a illustrates a domain shift problem where semantic categories like dog, horse, giraffe, and house appear across multiple domains such as art, cartoons, photos, and sketches. Although the label space is unchanged, visual attributes such as texture, color, and abstraction may vary significantly. This highlights two critical chal-

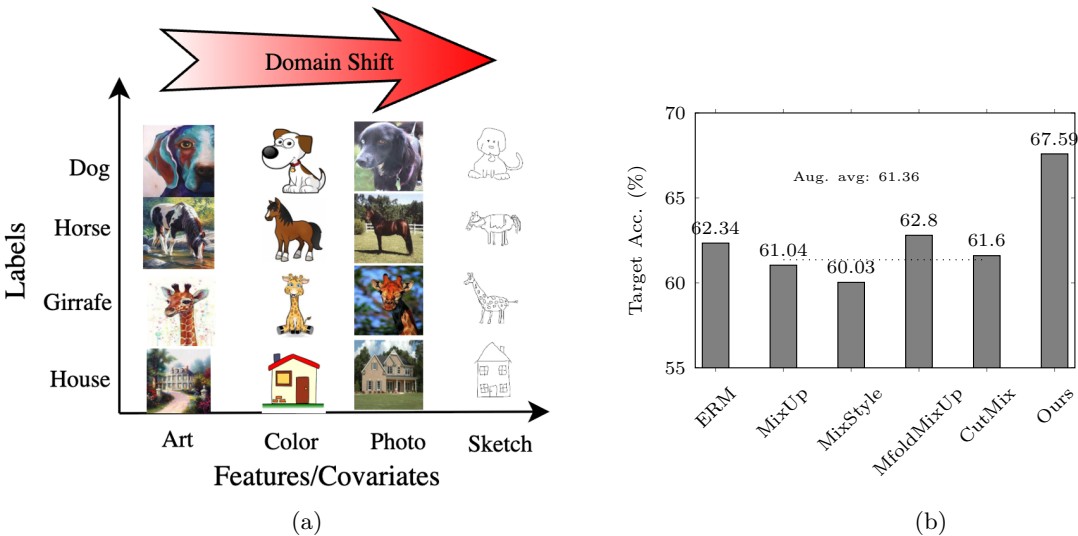

Figure 1: (a) Domain shift illustration. (b) DG target accuracy comparison averaged across five benchmarks.

lenges: (i) models trained on a single domain often fail to generalize to others, and (ii) successful generalization requires learning representations that ignore domain-specific cues while preserving label-discriminative features. Domain Generalization (DG) (Zhou et al., 2022) aims to address these challenges by leveraging multiple source domains to learn representations that remain stable across changes in the environment. The goal is to reduce reliance on domain-specific cues and instead capture features that are both label-discriminative and domain-invariant, enabling robust generalization performance across new and previously unseen domains.

Empirical Risk Minimization (ERM) (Vapnik, 1998) is considered a decent baseline method to achieve reasonable In-Distribution (ID) performance. However, without proper regularization, ERM tends to overfit to domain-specific spurious correlations, leading to poor generalization to unseen domains where such correlations may not hold. Another prevalent method in DG is achieved through data augmentation and synthesis (Li et al., 2018a; Zhou et al., 2020). These approaches perform augmentation by interpolating data points and their labels in input space (Zhang et al., 2017), hidden manifold space (Verma et al., 2019), and by mixing style statistical informations (eg: mean, std) of features (Zhou et al., 2021), or replacing random patches of one input with another while mixing their corresponding label (Yun et al., 2019). Such techniques are known to expose the model to a large variety of source domain samples, intrinsically learning more domain-invariant and hence generalizable representations. However, these methods are also prone to consuming a generous amount of computing resources and require considerable engineering efforts to synthesize inputs to represent domain-agnostic features. Besides, some of these methods utilize conventional augmentation methods (crop, rescale, rotate) and patch-based synthesis, limiting their versatility across other applications.

In this paper, we propose an alternative and relatively straightforward approach to enhance DG performance by regularizing the model to promote diverse feature learning. Our methodology comprises two steps: First, we calculate entropy to provide an uncertainty measure of the input data. Second, we utilize this entropy as supplementary signals to sample diverse features through Determinantal Point Process (DPP) sampling (Kulesza et al., 2012). DPP offers a class of precise probabilistic models for sample selection problems and is particularly effective for addressing subset selection problems with diversity constraints, such as video and document summarization (Launay et al., 2021). Our DPP sampling-guided regularization supports the original training objective of the existing ERM baseline to learn a domain-agnostic classifier. Unlike conventional data augmentation methods that achieve diversity by generating or synthesizing additional samples, our method inherently incorporates diversity within the sampling process.

Our main **contributions** are summarized as follows:

- We propose a DG method that enhances generalization across unseen domains by sampling diverse data representations near the decision boundary using uncertainty-guided DPP sampling.

- Our approach provides an effective form of regularization that reduces reliance on complex data augmentation or synthesis, offering a simpler alternative to augmentation-driven DG methods.

- We demonstrate that our regularization technique is **method-agnostic** and can be seamlessly integrated into existing DG frameworks by augmenting their final objectives, thereby promoting its applicability across a wide range of DG problems.

- Through empirical results, we show consistent improvements: (see Figure 1b) about **+5% over the ERM baseline, +6% over the average of popular augmentation-based DG benchmarks**. We also beat several state-of-the-art DG techniques across multiple datasets.

## 2    Background and Related Works

**Domain Generalization:** DG (Ye et al., 2022; Xu et al., 2021; Balaji et al., 2018) has been a subject of extensive investigation, alongside related approaches such as transfer learning and domain adaptation. A typical approach in DG is through learning domain-invariant representations across multiple domains (Li et al., 2018a; Muandet et al., 2013) with an assumption that all domains share common features that are inherently agnostic and do not belong to a particular domain. Existing methods attempt to learn these features through techniques such as invariant feature learning (Li et al., 2018b), disentanglement learning (Zhang et al., 2022), causal inference (Mahajan et al., 2021), etc. Although the distillation of domain-agnostic features for classification remains promising, the precise determination of which features should be considered domain-specific remains a significant question. Similarly, a substantial portion of the literature in DG has been realized through adversarial learning such as GANs (Poursaeed et al., 2021), and AutoEncoders (Ghifary et al., 2015) with an objective of learning universal representations. However, the majority of these aforementioned methodologies are computationally expensive and involve complex conceptualizations. In fact, recent research (Gulrajani & Lopez-Paz, 2020) challenges the performance of most of these models through a simple ERM framework that has undergone careful hyper-parameter tuning. Interestingly, it has been empirically demonstrated that a simple model such as ERM can achieve comparable results or even outperform complex approaches when evaluated in terms of OOD accuracy. Later studies (Cha et al., 2021; Wang et al., 2023) have argued that, simply minimizing the empirical loss on a complex and non-convex loss landscape is typically not sufficient to arrive at a good generalization, thereby proposing to find flat minima on the loss landscape for improving DG performance. Despite the improvement, such a gain is usually attributed to heuristic approximations and these methods fail to strongly utilize domain-agnostic information.

**Determinantal Point Process (DPP):** DPP has proven to be an effective methodology for ensuring diversity in a wide range of applications. Historically, DPPs have found favor in information retrieval tasks, including text summarization (Perez-Beltrachini & Lapata, 2021), image selection (Kulesza & Taskar, 2011), recommendation systems (Chen et al., 2018; Celis et al., 2018), and video summarization (Wilhelm et al., 2018). Beyond these fields, DPP sampling has demonstrated utility in various computer vision challenges including pose estimation (Kulesza & Taskar, 2010), image processing for pixels or patches sampling (Launay et al., 2021), transfer learning for generating diverse training subsets with enhanced transferability (Lv et al., 2022), multi-label classification (Xie et al., 2017), and active learning (Biyik et al., 2019). DOMI (Leng et al., 2022), which is one of the closest work to ours uses DPP as a two level sampling process however, it necessitates features from two distinct networks and offers little to no information regarding scalability to current DG benchmarks. Conversely, our approach can be seamlessly integrated as a straightforward plug-and-play technique with widely used existing methods, and we demonstrate its scalability across the current popular DG benchmarks.

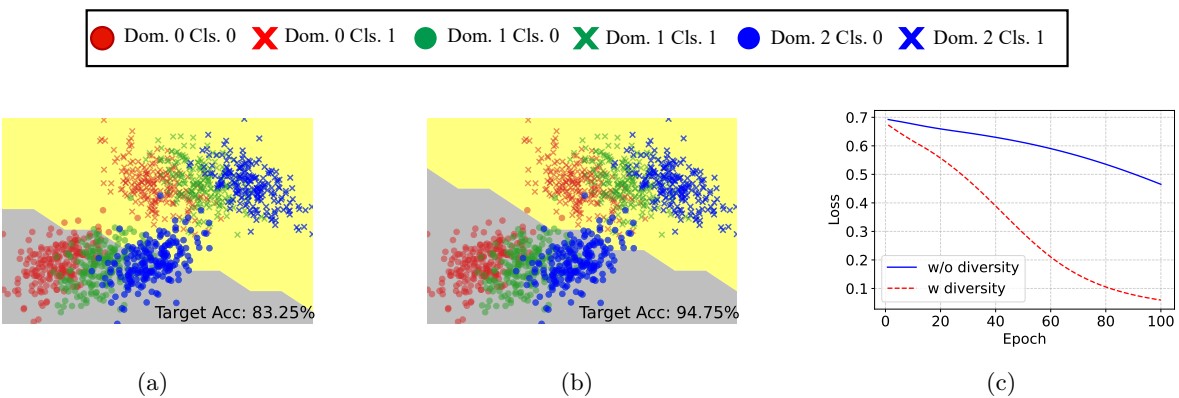

Figure 2: Comparison of decision boundaries for (a) ERM and (b) ERM with diversity regularization. ERM alone produces a misaligned boundary with lower target accuracy (83.25%). Adding the DPP-based diversity component yields a more robust boundary and substantially higher accuracy (94.75%), illustrating how diversity promotes domain-agnostic generalization. Source domains are [`Dom 0`, `Dom 1`], target domain `Dom 2`, with labels [`Cls 0`, `Cls 1`]. (c) Comparison of the classification loss curve relative to the number of epochs.

## 3  The Proposed Methodology

### 3.1  Problem Formulation

Let $\mathcal{X}$ denote the input space and $\mathcal{Y} = \{1, \ldots, C\}$ the label space for a $C$ class classification problem. We are given labeled samples from $N$ source domains $\mathcal{D}_1, \ldots, \mathcal{D}_N$, where each domain $\mathcal{D}_i$ contains samples $(\mathbf{x}, y) \sim \mathcal{P}_i(\mathcal{X}, \mathcal{Y})$. We consider a hypothesis class $\mathcal{F}$ defined as $\mathcal{F} = \{f_\theta \mid f_\theta(\mathbf{x}) = g(h(\mathbf{x})), \ h \in \mathcal{H}, \ g \in \mathcal{G}\}$, where $h : \mathcal{X} \to \mathbb{R}^d$ is a feature extractor and $g : \mathbb{R}^d \to \mathbb{R}^C$ is a classifier. The goal is to learn a model $f_\theta^* : \mathcal{X} \to \mathbb{R}^C$ that generalizes well on a previously unseen target domain $\mathcal{D}_T \sim \mathcal{P}_T(X, Y)$, where $\mathcal{P}_T \notin \{\mathcal{P}_i\}_{i=1}^N$. In other words, the model is trained to minimize the empirical risk on source domains $R_S(f) = \frac{1}{N} \sum_{i=1}^N \mathbb{E}_{(\mathbf{x},y) \sim \mathcal{D}_i}[\ell(f(\mathbf{x}), y)]$ to obtain:

$$f_\theta^* = \arg\min_{f \in \mathcal{F}} R_S(f) \tag{1}$$

with an assumption that $f_\theta^*$ properly generalizes to the target domain with true target risk given by:

$$R_T(f) = \mathbb{E}_{(\mathbf{x},y) \sim \mathcal{D}_T}[\ell(f(\mathbf{x}), y)]. \tag{2}$$

Then, under the standard domain shift assumption, the target risk can be upper bounded as:

$$R_T(f) \ \leq \ R_S(f) \ + \ \mathcal{C}\big(\{\mathcal{D}_i\}_{i=1}^N, \mathcal{D}_T\big), \forall f \in \mathcal{F} \tag{3}$$

Here, $\mathcal{C}$ denotes a generic domain discrepancy term that measures the distributional shift between the collection of source domains and the target domain, and can be instantiated by any divergence or distance for which a risk transfer bound of the form in (3) holds (e.g., integral probability metrics (IPM) or $f$ divergence based measures), depending on the underlying assumptions on the data distributions, loss function, and hypothesis class. Note that, throughout this work, we treat $\mathcal{C}$ abstractly and do not assume a specific form, as our method aims to reduce domain discrepancy implicitly via representation diversity rather than optimizing an explicit divergence.

### 3.2  Motivating Toy Example

We begin by motivating our problem through a simple toy example. Here, we construct a synthetic dataset using multivariate Gaussian distributions that comprises three domains and two classes; each class containing

200 data points. The dataset was crafted to maintain consistent covariance across all domains while altering only the mean, hence emulating the domain shift nature in a real-world DG task, where an object's feature values change based on the environmental context. Figure 2a shows the decision boundary learned by a standard ERM model trained across two source domains. Here's what we observe; although the model captures some of the class separation, the boundary is misaligned with the data distribution. As a result, substantial portion of the clusters of the target domain data are incorrectly classified, yielding a target accuracy of only 83.25%. This reflects a common limitation of ERM: the model tends to overfit to dominant patterns in the training set and fails to generalize well when class and domain structures overlap.

To this end, we introduce a diversity regularizer $\mathcal{R}_{\text{div}}$, which we integrate into the source risk minimization framework Eq. 1. Specifically, instead of minimizing only the empirical source risk $R_S(f)$, we consider the following objective:

$$f_\theta^* = \arg\min_{f \in \mathcal{F}} \left( R_S(f) + \mathcal{R}_{\text{div}} \right), \tag{4}$$

where, $\mathcal{R}_{\text{div}}$ is the diversity regularizer. By explicitly encouraging the model to attend to diverse samples during training, the decision boundary becomes more robust and better aligned with the data manifold as shown in Figure 2b. This adjustment resolves much of the mis-classification seen in the ERM, especially around the overlapping regions of different domains. Consequently, the target accuracy increases substantially to 94.75%. Building on this motivating example, we formalize our method **D**iversity **S**ampling **R**egularization (DSR). As shown in the block diagram in Figure 3a, DSR leverages predictive uncertainty followed by DPP sampling to induce diversity constraint among feature vectors $\mathbf{z}$, where $\mathbf{z} = h(\mathbf{x}) \in \mathbb{R}^d$. In the following sections, we outline how DSR is incorporated into the optimization objective alongside the source risk $R_S(f)$.

### 3.3 Diversity in Feature Space

There are two key arguments on how DPP supports to domain invariant feature learning. Firstly, DPP encourages inter- and intra-domain diversity, ensuring that the selected examples span a larger volume in feature space, covering more modes of the data distribution. Secondly, unlike random sampling, there is less redundancy in selected features, which makes the probability of selecting the same or similar samples very low. This can be attributed to the pivotal characteristic of DPP, which is a negative correlation between binary variables, translating that the selection of one item reduces the probability of selecting others that are highly similar. As shown in Figure 3b, the larger the angular spread, the more diverse the items are. We utilize these correlations, which are determined by a kernel matrix measuring item similarity. Consequently, more similar items are less likely to co-occur, resulting in DPPs favoring diverse subset of items.

### 3.4 Uncertainty-based Feature Modulation

In DG, not all samples contribute equally to improving generalization. In reality, confident predictions often correspond to redundant or well-understood features, whereas uncertain predictions may represent underexplored regions of the input space (Hüllermeier & Waegeman, 2021). To exploit this, we compute the predictive entropy $\mathbf{u} = -\sum_{c=1}^{C} \hat{y}_c \log y_c$ for each sample based on the softmax output of the classifier $\hat{y}_c = \text{softmax}(g(h(\mathbf{x})))$. This entropy is then used to scale each feature vector, effectively reweighting the feature matrix before passing for DPP sampling. We term this phenomenon as feature modulation and represent the byproduct as modulated feature given by:

$$\tilde{\mathbf{z}} = \mathbf{z} \cdot \mathbf{u} \tag{5}$$

The intuition here is that uncertain samples are likely to contain more informative or ambiguous signals, and emphasizing them encourages the model to explore diverse and non-redundant directions in feature space. This entropy-based modulation serves two primary purposes. First, it biases learning toward instances the model already deems informative, implicitly encouraging robustness across domains. Second, and more critically for our method, it alters the structure of the feature space on which we perform DPP-based diversity sampling. Empirically, we find that this strategy leads to more expressive representations and contributes significantly to performance under domain shift.

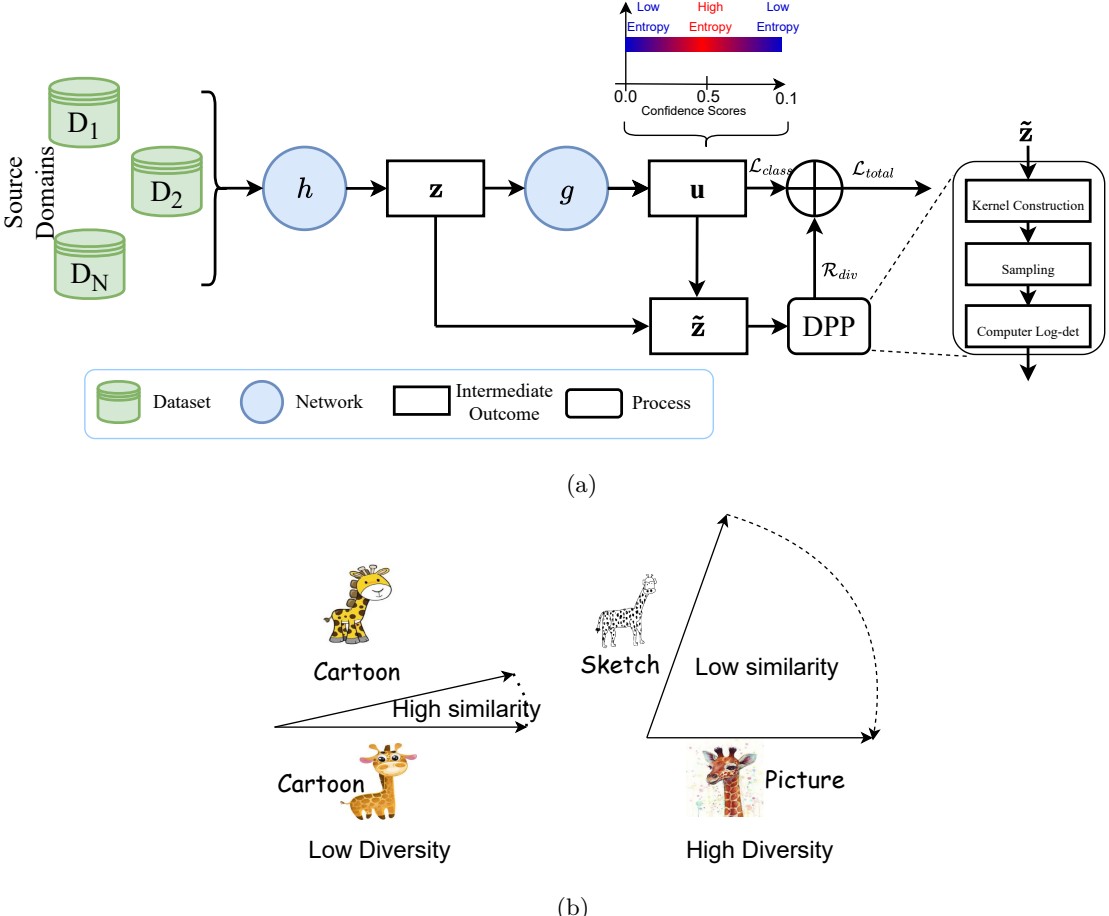

(a)

(b)

Figure 3: (a) Schematic block diagram of DSR method. Based on the entropy scale, the high entropy values are prioritized for DPP sampling. (b) Diversity representation between data points. The larger the angular spread, the higher the diversity.

### 3.5 Diversity Kernel with DPP

Next, to promote diverse representation during training, we construct a kernel matrix $\mathbf{L} \in \mathbb{R}^{B \times B}$ over the batch, $B$ of the modulated feature vectors $\tilde{\mathbf{z}}$ using an RBF (Gaussian) similarity kernel.

$$\mathbf{L}_{ij} = \exp(-\gamma ||\tilde{\mathbf{z}}_i - \tilde{\mathbf{z}}_j||_2^2) \tag{6}$$

This kernel captures pairwise dissimilarities among feature vectors. In situations where two features are exactly same, $\tilde{\mathbf{z}}_i = \tilde{\mathbf{z}}_j$, $\mathbf{L}_{ij} \to 1$. However, two independent features in a batch are never exactly the same, given they come from different input samples, which brings our $\mathbf{L}_{ij}$ to span between 0 and 1.

**Role of $\gamma$:** As the inverse bandwidth parameter, $\gamma > 0$ determines the width of the RBF kernel. Since the mini-batch comprises samples from different domains that are shifted along the covariates, we use a data-dependent $\gamma$ which can capture the variances that are intrinsic to the shift dynamics. Specifically, we use a principled, data-dependent median heuristic measure (Garreau et al., 2017; Gretton et al., 2012) that is computed over pairwise distances of features in a batch. Intuitively, the median value renders $\gamma$ to be robust to outliers and automatically adapts to the scale of data in inadequately shifted distribution space. Mathematically, it is represented as follows:

$$\gamma = \frac{1}{\text{median}||\tilde{\mathbf{z}}_i - \tilde{\mathbf{z}}_j||_2^2 + \epsilon} \tag{7}$$

where $\epsilon$ is a small positive constant that ensures numerical stability. Since the modulated features reflect uncertainty, the diversity modeled by the DPP naturally emphasizes informative regions of the feature space. This leads to more meaningful subset selection and loss computation, eventually improving generalization.

### 3.6 Reducing Domain Discrepancy through Diversity Regularizer $\mathcal{R}_{\text{div}}$

Next, we utilize the previously obtained $\mathbf{L}$, to enforce diversity in the learned feature representations via a log-determinant regularizer:

$$\mathcal{R}_{\text{div}} = -\log\det(\mathbf{L} + \epsilon I), \tag{8}$$

To ensure that the kernel matrix $\mathbf{L}$ is strictly positive definite and the log-determinant is well-defined, we regularize it by adding a small multiple of the identity matrix $I$, with $\epsilon > 0$ being a small constant. This guarantees that all eigenvalues of $\mathbf{L}$ remain positive, avoiding numerical instability caused by near-zero or negative eigenvalues. Such a regularization is commonly practiced in determinant-based objectives and ensures that the $\log\det$ term remains finite and differentiable (Kulesza et al., 2012). This regularization encourages the feature vectors to be geometrically diverse, suppressing collapse into redundant subspaces.

**Proposition 1.** *Maximizing* $\log\det(\mathbf{L} + \epsilon I)$ *promotes non-redundant, orthogonal features that span a high-volume subspace, thereby improving representation robustness.*

**Lemma 1.** *(Kulesza et al., 2012) Let* $\lambda_1, \ldots, \lambda_m$ *be the eigenvalues of* $\mathbf{L}$. *Then,*

$$\log\det(\mathbf{L} + \epsilon I) = \sum_{i=1}^{m} \log(\lambda_i + \epsilon),$$

*which is maximized when the eigenvalues are uniformly large and spread out, implying orthogonality and high entropy in the feature distribution.*

**Proof sketch:** Since $\det(\mathbf{L} + \epsilon I) = \prod_{i=1}^{m}(\lambda_i + \epsilon)$, maximizing its logarithm requires enlarging all eigenvalues of $\mathbf{L}$. If the spectrum collapses so that only a few eigenvalues dominate, the determinant shrinks, reflecting concentration of variance in a low-dimensional subspace. Conversely, when eigenvalues are more uniformly distributed and nonzero, the determinant grows, corresponding to orthogonal features that span a higher-volume subspace. Thus, maximizing $\log\det(\mathbf{L} + \epsilon I)$ promotes diversity by penalizing redundancy among features.

**Discussion:** Diverse representations reduce the overlap between features of different domains, leading to more domain-invariant embeddings. Maximizing diversity thus implicitly aligns source domains in a shared high-volume subspace, shrinking the discrepancy term $\mathcal{C}$ in the generalization bound in Eq. 3.

### 3.7 Interpretation of the log-det

In DPP sampling, the model is penalized if the mini-batch samples are redundant or lie in a low-dimensional subspace. To illustrate the log-det procedure mathematically, let us consider the subset $S$ containing two data points $\mathbf{x}_i$, and $\mathbf{x}_j$, then the probability of sampling can be calculated as:

$$\begin{aligned}
\mathcal{P}(i, j \in S) \propto \det(\mathbf{L}_S) &= \begin{vmatrix} \mathbf{L}_{ii} & \mathbf{L}_{ij} \\ \mathbf{L}_{ji} & \mathbf{L}_{jj} \end{vmatrix} \\
&= \mathbf{L}_{ii}\mathbf{L}_{jj} - \mathbf{L}_{ij}\mathbf{L}_{ji} = 1 - \mathbf{L}_{ij}^2
\end{aligned} \tag{9}$$

From Eqs. 6, and 9, we can deduce that, under ideal scenario if $\mathbf{x}_i$, and $\mathbf{x}_j$ are dissimilar to each other, then $\mathbf{L}_{ij} \to 0$, and $S$ will have high probability. Accordingly, the selected subset with the highest $\mathcal{P}$ should contain a set of highly diverse samples.

It is important to note that in geometric intuition, the determinant $\det(\mathbf{L}_S)$ represents volume of the parallelepiped spanned by the feature vectors $\tilde{\mathbf{z}} \in S$ in the feature space [1]. Thus, by maximizing $\det(\mathbf{L}_S)$, the selected subset $S$ ensures that the feature vectors are linearly independent and spread out in representation space. This diversity implies that our method captures a wide variety of diverse features among $\tilde{\mathbf{z}}$, reducing the likelihood that the model overfits to any specific domain.

## 3.8 Overall Training Objective

With the regularization term formulated, DSR's final training objective corresponding to Eq. 4 becomes as follows: $\mathcal{L}_{\text{total}} = \alpha \cdot \underbrace{\frac{1}{N} \sum_{i=1}^{N} \mathbb{E}_{(\mathbf{x},y) \sim \mathcal{D}_i} [\ell(f(\mathbf{x}), y)]}_{\text{ERM}} + (1 - \alpha) \cdot \mathcal{R}_{\text{div}}$ where $\alpha \in [0, 1]$ balances the contribution of the diversity term.

# 4 Experimental Setup

In this section, we outline the experimental setup of our method. This includes details regarding the datasets used, training configuration, evaluation metrics and selected baseline studies for comparison. All experiments were conducted on multiple A100 GPU servers.

## 4.1 Datasets

**PACS:** The PACS dataset (Li et al., 2017) comprises four distinct domains: Photo, Art Painting, Cartoon, and Sketch. Each domain contains images categorized into seven classes: Dog, Elephant, Giraffe, Guitar, House, Horse, and Person.

**VLCS:** The VLCS dataset (Fang et al., 2013) combines four distinct image datasets: PASCAL VOC 2007 (V), LabelMe (L), Caltech-101 (C), and SUN09 (S). This consolidated dataset encompass five shared categories: Bird, Car, Chair, Dog, and Person.

**OfficeHome (OH):** The OfficeHome dataset (Venkateswara et al., 2017), a multi-domain dataset, was developed to facilitate research in domain adaptation and generalization. This dataset encompasses four distinct domains: Art, Clipart, Product, and Real-World.

**TerraIncognita (TI):** TerraIncognita (Beery et al., 2018) comprises wildlife camera trap images collected from multiple geographical regions, referred to as "locations."

**DomainNet (DN):** DomainNet (Peng et al., 2019) is a prominent dataset in DG, comprising six distinct domains: Clipart, Infograph, Painting, Quickdraw, Real, and Sketch.

In Table. 1, we provide detailed information regarding the distribution of samples in each domain of the DG datasets as discussed above. We can observe that the number of image samples in each domain vary significantly across all datasets.

## 4.2 Implementation Details

We trained our method on Resnet-50 architecture using the Stochastic Gradient Descent (SGD) optimizer for up to 100 epochs. The learning rate was set to 0.01 with a decay rate of 0.75. The value of batch size was set to 64 and the value of $\alpha$ was set to 0.5 to balance the contributions of the loss term and the regularizer. We follow the training-validation approach of (Wang & Lu, 2021) where any one domain in the dataset is reserved for testing purposes while the rest are used for training. Our method's performance is compared against some of the widely recognized DG baselines: ERM (Vapnik, 1998), Mixup (Yan et al., 2020), CORAL (Sun & Saenko, 2016), MMD (Li et al., 2018b), DANN (Li et al., 2018b), GroupDRO (Sagawa et al., 2019),

---

[1]In DPP, the determinant is viewed as an area of the parallelogram induced when a unit square is transformed by a matrix.

Table 1: Summary of DG Benchmarks Used in the Experiments.

| Dataset | Domain | Images | Total |
|---|---|---|---|
| PACS | Art Painting | 2,048 | 9,991 |
| | Cartoon | 2,344 | |
| | Photo | 1,670 | |
| | Sketch | 3,929 | |
| VLCS | Caltech101 | 1,415 | 10,729 |
| | LabelMe | 2,656 | |
| | SUN09 | 3,282 | |
| | VOC2007 | 3,376 | |
| OfficeHome | Art | 2,427 | 15,588 |
| | Clipart | 4,365 | |
| | Product | 4,439 | |
| | Real-World | 4,357 | |
| TerraIncognita | Location 100 | 3,381 | 24,788 |
| | Location 38 | 9,530 | |
| | Location 43 | 6,293 | |
| | Location 46 | 5,584 | |
| DomainNet | Clipart | 48,129 | 586,575 |
| | Infograph | 53,201 | |
| | Painting | 72,759 | |
| | Quickdraw | 172,500 | |
| | Real | 175,327 | |
| | Sketch | 70,386 | |

and S.O.T.A methods such as SWAD (Cha et al., 2021), SAGM (Wang et al., 2023), GMDG (Tan et al., 2024), and I3C (Zhou et al., 2025). We re-implement most of the methods and report the mean and standard error across three random seeds.

| METHODS | PACS | VLCS | OH | TI | DN | Avg. |
|---|---|---|---|---|---|---|
| ERM (Vapnik, 1998) | $84.90 \pm 0.11$ | $74.20 \pm 0.15$ | $67.30 \pm 0.07$ | $45.50 \pm 0.10$ | $39.80 \pm 0.33$ | 62.34 |
| MixUp (Yan et al., 2020) | $79.50 \pm 0.14$ | $73.90 \pm 0.18$ | $66.60 \pm 0.21$ | $47.30 \pm 0.17$ | $37.90 \pm 0.02$ | 61.04 |
| CORAL (Sun & Saenko, 2016) | $82.50 \pm 0.19$ | $74.30 \pm 0.57$ | $66.80 \pm 0.20$ | $39.50 \pm 0.47$ | $36.40 \pm 0.49$ | 59.9 |
| GroupDRO (Sagawa et al., 2019) | $82.60 \pm 0.08$ | $71.10 \pm 0.24$ | $63.00 \pm 0.40$ | $39.40 \pm 0.24$ | $32.40 \pm 0.18$ | 57.7 |
| MMD (Li et al., 2018b) | $84.70 \pm 0.17$ | $75.70 \pm 0.06$ | $67.80 \pm 0.19$ | $41.80 \pm 0.20$ | $38.50 \pm 0.40$ | 61.7 |
| SWAD (Cha et al., 2021) | $\underline{88.19} \pm 0.25$ | $78.77 \pm 0.16$ | $\mathbf{70.50} \pm \mathbf{0.13}$ | $47.50 \pm 0.19$ | $\mathbf{46.40} \pm \mathbf{0.13}$ | $\underline{66.27}$ |
| SAGM (Wang et al., 2023) | $86.72 \pm 0.22$ | $77.13 \pm 0.14$ | $69.26 \pm 0.58$ | $48.43 \pm 0.51$ | $44.60 \pm 0.37$ | 65.22 |
| GMDG† (Tan et al., 2024) | 85.60 | 79.20 | 70.70 | 50.20 | 44.60 | 66.06 |
| I3C† (Zhou et al., 2025) | 87.10 | 79.60 | 70.20 | 49.60 | 45.60 | 66.42 |
| $DSR_z$ (Ours) | $88.12 \pm 0.31$ | $\underline{79.10} \pm 0.41$ | $70.00 \pm 0.14$ | $\underline{51.41} \pm 0.73$ | $\underline{45.60} \pm 0.27$ | $\mathbf{67.54}$ |
| $DSR_{\tilde{z}}$ (Ours) | $\mathbf{88.70} \pm \mathbf{0.09}$ | $\mathbf{80.31} \pm \mathbf{0.14}$ | $\underline{70.24} \pm 0.31$ | $\mathbf{52.93} \pm \mathbf{0.47}$ | $\underline{45.75} \pm 0.29$ | $\mathbf{67.59}$ |

Table 2: DG target accuracy comparison of DSR along with popular baselines across five different benchmark datasets. Best and second-best results are bold and underlined, respectively. The results of methods with † are reported from their original implementation, therefore we do not provide the standard error value for them.

# 5 Experimental Evaluation

In this section, we conduct extensive experiments and analyses to evaluate the effectiveness of the proposed method on standard DG benchmarks and to provide extensive insights into its associated parameters. For evaluation, we use target accuracy, which is obtained by testing the model on the target domain that has no sample overlap with the source (training) domains. Note that the results are averaged over 3 independent trials, and represent the mean value and standard error across these runs.

| METHODS | Art | Color | Picture | Sketch | Avg. | Caltech101 | LabelMe | SUN09 | VOC2007 | Avg. |
|---|---|---|---|---|---|---|---|---|---|---|
| ERM | $84.10 \pm 0.23$ | $80.50 \pm 0.02$ | $94.90 \pm 0.08$ | $80.20 \pm 0.11$ | 84.90 | $96.40 \pm 0.10$ | $62.70 \pm 0.16$ | $68.40 \pm 0.31$ | $69.20 \pm 0.03$ | 74.20 |
| MixUp | $85.10 \pm 0.12$ | $80.90 \pm 0.17$ | $94.70 \pm 0.20$ | $57.30 \pm 0.09$ | 79.50 | $94.60 \pm 0.21$ | $62.50 \pm 0.43$ | $69.50 \pm 0.08$ | $69.00 \pm 0.01$ | 73.90 |
| CORAL | $82.30 \pm 0.25$ | $80.20 \pm 0.13$ | $92.80 \pm 0.06$ | $74.90 \pm 0.34$ | 82.50 | $94.60 \pm 0.76$ | $62.00 \pm 0.52$ | $71.50 \pm 0.30$ | $69.20 \pm 0.71$ | 74.30 |
| GroupDRO | $76.70 \pm 0.20$ | $78.60 \pm 0.10$ | $90.00 \pm 0.03$ | $85.00 \pm 0.00$ | 82.60 | $93.90 \pm 0.61$ | $60.40 \pm 0.30$ | $65.90 \pm 0.27$ | $64.20 \pm 0.21$ | 71.10 |
| MMD | $83.50 \pm 0.21$ | $80.10 \pm 0.32$ | $93.50 \pm 0.08$ | $81.80 \pm 0.09$ | 84.70 | $96.30 \pm 0.02$ | $61.70 \pm 0.03$ | $73.80 \pm 0.11$ | $70.90 \pm 0.08$ | 75.70 |
| SWAD | $\underline{88.59} \pm 0.14$ | $83.58 \pm 0.51$ | $\mathbf{97.82} \pm 0.22$ | $82.76 \pm 0.16$ | $\underline{88.19}$ | $\mathbf{98.94} \pm 0.14$ | $63.71 \pm 0.11$ | $74.10 \pm 0.20$ | $\underline{78.35} \pm 0.22$ | 78.77 |
| SAGM | $86.51 \pm 0.33$ | $83.74 \pm 0.17$ | $96.70 \pm 0.27$ | $79.93 \pm 0.13$ | 86.72 | $98.58 \pm 0.19$ | $61.27 \pm 0.09$ | $72.31 \pm 0.13$ | $76.37 \pm 0.15$ | 77.13 |
| GMDG | 84.7 | 81.7 | 97.50 | 80.50 | 85.6 | 98.30 | 65.90 | 73.40 | 79.30 | 79.20 |
| I3C | 89.20 | 80.60 | 97.70 | 81.40 | 87.10 | 98.80 | 67.50 | 72.10 | 80.20 | $\underline{79.60}$ |
| $\text{DSR}_{\mathbf{z}}$ | $87.74 \pm 0.23$ | $\underline{83.75} \pm 0.12$ | $97.02 \pm 0.51$ | $\underline{83.97} \pm 0.22$ | 88.12 | $98.09 \pm 0.10$ | $\underline{67.42} \pm 0.33$ | $\underline{74.63} \pm 0.79$ | $76.26 \pm 0.43$ | 79.10 |
| $\text{DSR}_{\tilde{\mathbf{z}}}$ | $\mathbf{89.26} \pm 0.09$ | $\mathbf{84.04} \pm 0.02$ | $\underline{97.25} \pm 0.12$ | $\mathbf{84.25} \pm 0.13$ | 88.70 | $\underline{98.26} \pm 0.18$ | $\mathbf{68.90} \pm 0.12$ | $\mathbf{75.69} \pm 0.15$ | $78.41 \pm 0.11$ | 80.31 |

(a) PACS and VLCS

| METHODS | Art | Clipart | Product | Real | Avg. | Loc38 | Loc43 | Loc46 | Loc100 | Avg. |
|---|---|---|---|---|---|---|---|---|---|---|
| ERM | $62.20 \pm 0.04$ | $53.80 \pm 0.08$ | $76.00 \pm 0.10$ | $77.40 \pm 0.09$ | 67.30 | $46.10 \pm 0.11$ | $53.40 \pm 0.12$ | $41.90 \pm 0.13$ | $40.80 \pm 0.06$ | 45.50 |
| MixUp | $60.30 \pm 0.22$ | $53.50 \pm 0.20$ | $75.90 \pm 0.31$ | $76.60 \pm 0.14$ | 66.60 | $44.30 \pm 0.18$ | $52.50 \pm 0.19$ | $33.50 \pm 0.09$ | $\underline{58.90} \pm 0.23$ | 47.30 |
| CORAL | $59.60 \pm 0.21$ | $55.90 \pm 0.26$ | $75.00 \pm 0.19$ | $76.80 \pm 0.17$ | 66.80 | $39.20 \pm 0.37$ | $41.60 \pm 0.61$ | $32.20 \pm 0.51$ | $45.00 \pm 0.39$ | 39.50 |
| GroupDRO | $54.20 \pm 0.71$ | $53.30 \pm 0.12$ | $72.40 \pm 0.45$ | $72.00 \pm 0.34$ | 63.00 | $35.40 \pm 0.28$ | $42.00 \pm 0.21$ | $31.90 \pm 0.33$ | $48.20 \pm 0.16$ | 39.40 |
| MMD | $60.70 \pm 0.12$ | $\mathbf{58.70} \pm 0.16$ | $75.60 \pm 0.28$ | $76.30 \pm 0.21$ | 67.80 | $40.40 \pm 0.31$ | $47.10 \pm 0.18$ | $34.40 \pm 0.18$ | $45.10 \pm 0.14$ | 41.80 |
| SWAD | $65.24 \pm 0.03$ | $\underline{57.93} \pm 0.13$ | $\mathbf{78.99} \pm 0.17$ | $\mathbf{79.86} \pm 0.19$ | 70.50 | $36.74 \pm 0.21$ | $\mathbf{59.79} \pm 0.11$ | $37.93 \pm 0.19$ | $55.55 \pm 0.27$ | 47.50 |
| SAGM | $64.26 \pm 0.78$ | $55.92 \pm 0.45$ | $\underline{78.04} \pm 0.67$ | $78.83 \pm 0.43$ | 69.26 | $45.04 \pm 0.61$ | $\underline{57.96} \pm 0.32$ | $40.26 \pm 0.87$ | $50.48 \pm 0.26$ | 48.43 |
| GMDG | 68.90 | 56.20 | 79.90 | 82.00 | 70.70 | 59.80 | 45.30 | 57.10 | 38.20 | 50.20 |
| I3C | 67.50 | 56.10 | 77.50 | 79.80 | 70.20 | 57.00 | 43.20 | 56.40 | 41.90 | 49.60 |
| $\text{DSR}_{\mathbf{z}}$ | $\underline{66.05} \pm 0.16$ | $56.85 \pm 0.26$ | $77.40 \pm 0.11$ | $79.73 \pm 0.03$ | 70.00 | $\underline{48.22} \pm 0.84$ | $55.76 \pm 0.06$ | $\underline{43.63} \pm 0.81$ | $58.05 \pm 1.23$ | $\underline{51.41}$ |
| $\text{DSR}_{\tilde{\mathbf{z}}}$ | $\mathbf{66.19} \pm 0.23$ | $57.21 \pm 0.26$ | $77.61 \pm 0.34$ | $\underline{79.79} \pm 0.41$ | $\underline{70.20}$ | $\mathbf{50.63} \pm 0.59$ | $54.90 \pm 0.30$ | $\mathbf{45.61} \pm 0.41$ | $\mathbf{60.59} \pm 0.61$ | 52.93 |

(b) OfficeHome and TerraIncognita

| METHODS | Clipart | Infograph | Painting | Quickdraw | Real | Sketch | Avg. |
|---|---|---|---|---|---|---|---|
| ERM | $58.00 \pm 0.19$ | $16.70 \pm 0.21$ | $45.30 \pm 0.43$ | $14.20 \pm 0.34$ | $56.70 \pm 0.65$ | $48.10 \pm 0.18$ | 39.80 |
| MixUp | $54.90 \pm 0.34$ | $16.30 \pm 0.57$ | $43.70 \pm 0.49$ | $13.70 \pm 0.65$ | $52.70 \pm 0.89$ | $46.10 \pm 0.36$ | 37.90 |
| CORAL | $57.10 \pm 0.87$ | $14.60 \pm 0.37$ | $39.80 \pm 0.31$ | $10.70 \pm 0.75$ | $50.20 \pm 0.40$ | $46.40 \pm 0.29$ | 36.40 |
| GroupDRO | $46.30 \pm 0.12$ | $15.50 \pm 0.09$ | $34.60 \pm 0.17$ | $10.80 \pm 0.43$ | $46.10 \pm 0.09$ | $41.10 \pm 0.21$ | 32.40 |
| MMD | $57.90 \pm 0.23$ | $16.00 \pm 0.12$ | $41.70 \pm 0.34$ | $13.00 \pm 0.76$ | $54.30 \pm 0.61$ | $48.00 \pm 0.37$ | 38.50 |
| SWAD | $\mathbf{65.75} \pm 0.12$ | $\mathbf{22.42} \pm 0.07$ | $\mathbf{52.83} \pm 0.11$ | $\underline{15.57} \pm 0.09$ | $\mathbf{66.33} \pm 0.23$ | $\mathbf{55.52} \pm 0.16$ | $\mathbf{46.40}$ |
| SAGM | $64.13 \pm 0.31$ | $20.56 \pm 0.21$ | $51.16 \pm 0.56$ | $14.22 \pm 0.51$ | $64.06 \pm 0.39$ | $53.47 \pm 0.26$ | 44.60 |
| GMDG | 63.40 | 22.40 | 51.40 | 13.40 | 64.40 | 52.40 | 44.60 |
| I3C | 63.00 | 21.40 | 51.40 | 13.60 | 63.60 | 53.50 | 44.40 |
| $\text{DSR}_{\mathbf{z}}$ | $65.30 \pm 0.32$ | $22.00 \pm 0.27$ | $52.20 \pm 0.14$ | $15.30 \pm 0.09$ | $64.30 \pm 0.19$ | $54.10 \pm 0.61$ | 45.60 |
| $\text{DSR}_{\tilde{\mathbf{z}}}$ | $\underline{65.34} \pm 0.14$ | $\underline{22.40} \pm 0.33$ | $\underline{52.31} \pm 0.56$ | $\mathbf{15.70} \pm 0.21$ | $\underline{64.50} \pm 0.43$ | $\underline{54.27} \pm 0.06$ | $\underline{45.75}$ |

(c) DomainNet

Table 3: A comprehensive comparison of DG performance of different methods across multiple benchmark datasets Rows highlighted in blue represent our methods. Best and second-best results are bold and underlined respectively.

## 5.1 Performance Comparison

In Table 2, we compare the accuracies of the proposed method with popular DG benchmarks along with recent SOTA frameworks. Here, we present two variants of our method: $\text{DSR}_{\mathbf{z}}$, and $\text{DSR}_{\tilde{\mathbf{z}}}$. The former utilizes plain features, while the latter uses entropy modulated features for DPP sampling. We can see that through simple regularization, our method provides an impressive boost to the ERM baseline, with almost 5% overall gain across the average of five different datasets. Both $\text{DSR}_{\mathbf{z}}$, and $\text{DSR}_{\tilde{\mathbf{z}}}$ were able to outperform most of the methods, while only coming second to SAGM and SWAD by a marginal value across OH and DN benchmarks respectively. One thing to note here is that, SAGM and SWAD both are designed on complex heuristics of the loss landscapes and provide little information regarding the domain features. On the other hand, our method is complementary to the existing loss function and do not require modifying any of the original ERM objective. In fact, we show in later section that our approach is also agnostic to most of the models and can boost their accuracy by a comprehensive margin.

Another insight to take from Table 2, is that MixUp, a popular data augmentation technique known for promoting diversity in the representation space, performs considerably worse than both DSR variants. We further illustrate this difference through a t-SNE plot in Figure 4, which shows that DSR achieves more compact inter-class separation on the target dataset compared to ERM and MixUp. The comprehensive experimental results comparison of $\text{DSR}_{\mathbf{z}}$, and $\text{DSR}_{\tilde{\mathbf{z}}}$ with the baselines is provided in Table 3 where we can witness that, in some domains (Loc100 of TerraIncognita), DSR brings upto 50% boost in the accuracy

of vanilla ERM. These findings suggest that diversity-sampling regularization markedly improves the DG performance.

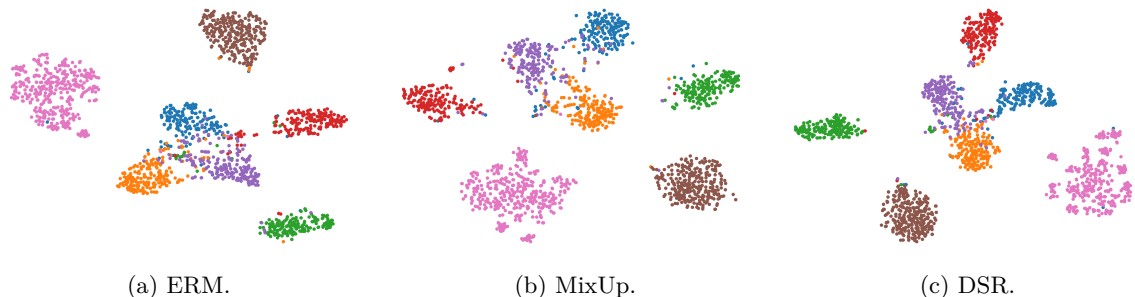

(a) ERM.                     (b) MixUp.                     (c) DSR.

Figure 4: TSNE plot of a target domain of the PACS dataset. Each colored cluster represents a class. (a) ERM (b) MixUp (c) DSR.

## 5.2 Model-Agnosticity of DSR

One of the key characteristics of DSR is that it is model agnostic, meaning that it can be coupled with any other method by augmenting their training objective. We show this phenomenon in Figure 5, where we plot the target accuracy of vanilla methods (ERM, DANN, CORAL, MMD, and GroupDRO) and their DSR counterparts averaged across all the available domains of five DG benchmark datasets as discussed in Section 4.1. Through the bar chart, it can be clearly seen that all methods have a certain degree of gain when they are trained with DSR. These gains starting from 1% reaching as high as up to 13% reflect the versatility of DSR, promoting its utility among wide range of methods and frameworks.

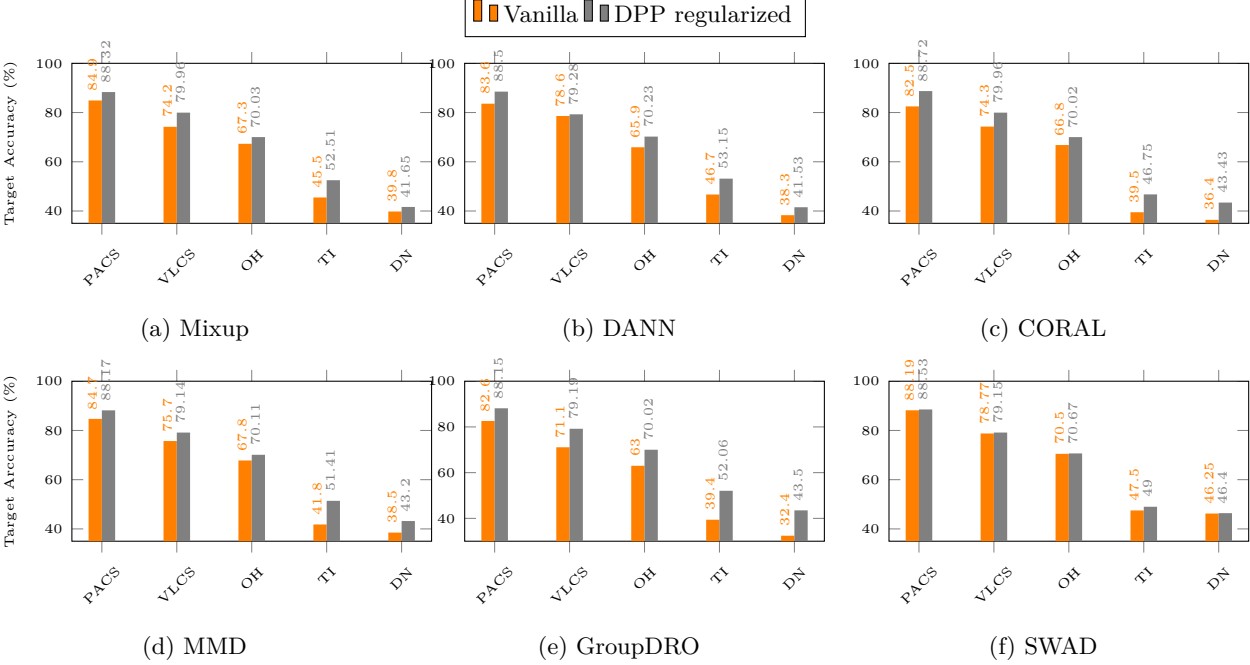

Figure 5: Comparison of target accuracy of five of the popular DG methods across five different benchmark datasets when trained on a vanilla and DSR regularized setting.

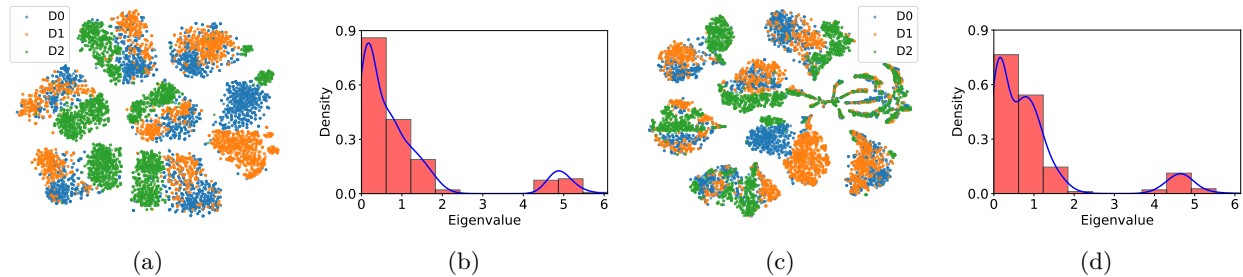

Figure 6: Analysis of features and eigenvalues after model training on the PACS dataset. (a) t-SNE plot with $\mathbf{z}$, (b) Histogram and KDE plot of eigen values using $\mathbf{z}$, (c) t-SNE plot with $\tilde{\mathbf{z}}$, (d) Histogram and KDE plot of eigen values using $\tilde{\mathbf{z}}$. In (a), and (c) each color represents a domain and each cluster represents a class.

## 5.3 Analysis on Feature Modulation of DSR

In Section 5.1, we presented two variants of DSR: $\mathrm{DSR}_\mathbf{z}$, and $\mathrm{DSR}_{\tilde{\mathbf{z}}}$: where the latter consistently provided better DG performance across all domains of all datasets. We attribute this improvement to an effective diversity regularizer driven by entropy-modulated features. In this section, by analyzing the feature values and distribution of kernel eigenvalues, we clarify why this improvement persists. To better understand the impact of entropy-based feature modulation, first we analyzed both the training feature space structures of $\mathbf{z}$ and $\tilde{\mathbf{z}}$ through t-SNE visualization as shown in Figures 6a, and 6c respectively. Here, we notice two key observations: First, $\tilde{\mathbf{z}}$ shows tightly clustered inter-class samples and the features appear more dispersed across the embedding space. Second, the clustered features appear more domain-agnostic, with the three domains converging toward a shared representational basis. From these findings, we deduce that modulation encourages the model to exploit more orthogonal directions in the feature space, thereby preventing redundancy in the learned representation.

Subsequently, we further substantiate this observation through an eigenvalue analysis of the kernel matrices constructed from these features. In Figures 6b, and 6d, we plot the density of eigenvalues obtained from these features. Here, we empirically validate Lemma 1, as the eigenvalue spectrum kernel matrix of $\tilde{\mathbf{z}}$ appear more spread out, with a lower proportion of negligible ($\approx 0$) eigenvalues and larger support volume in the feature space. Such a broader set of informative, non-redundant features benefits DPP sampling, reducing over-reliance on highly confident yet redundant features, thereby improving generalization in the context of domain shift.

## 5.4 Ablation Study: Effect of $\gamma$

In this ablation, we perform an experiment to assess the impact of using fixed $\gamma$ values on generalization performance. We trained $\mathrm{DSR}_{\tilde{\mathbf{z}}}$ with static $\gamma$ values from the set $\{0.01, 0.03, \dots, 0.1\}$, aiming to mirror the empirical range noted during adaptive training. We plot the line-graph of target accuracies with respect to $\gamma$ as shown in Figure 7a, where we also append the result of adaptive $\gamma$ at the end. Interestingly, we notice that not a single fixed value of $\gamma$ leads to optimal performance. This reveals that a fixed $\gamma$ in $\mathrm{DSR}_{\tilde{\mathbf{z}}}$ fails to capture the data-dependent nuances of the feature space. Conversely, the adaptive $\gamma$ designed to fit the statistical structure of the features at each training step, promotes effective diversity regularization yielding better target accuracy. These results support our assertion that adaptively computing $\gamma$, such as through the median of pairwise distances enhances DG performance.

### 5.4.1 Training on Different Alpha Values

Figure 7b illustrates the effect of varying the hyperparameter $\alpha$ on target-domain accuracy across the PACS, VLCS, and OH datasets. In the case of PACS, accuracy increases until $\alpha = 0.5$, where it attains a peak of 88.7%, before experiencing a slight decline at higher values. A similar pattern is observed for VLCS, with performance peaking at $\alpha = 0.5$ (80.5%) and remaining relatively stable at other values. The OH

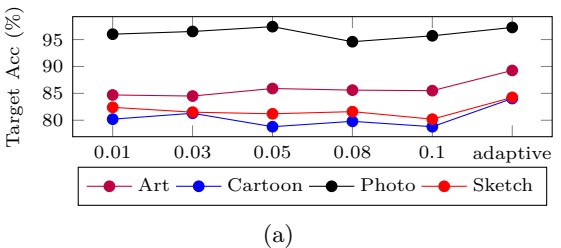 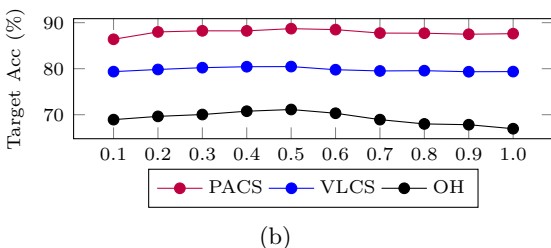

(a)             (b)

Figure 7: (a) Target accuracy on PACS dataset with static and adaptive $\gamma$. (b) Ablation on alpha values for PACS dataset.

dataset also demonstrates sensitivity to $\alpha$, achieving its maximum accuracy of 71.1% at $\alpha = 0.5$, followed by a decrease as $\alpha$ increases. Overall, the findings suggest that moderate values of $\alpha$ (approximately 0.5) consistently result in optimal performance across all datasets.

### 5.4.2   Training with Different Kernels

Figure 8a illustrates a kernel ablation study conducted on the PACS, VLCS, and OH datasets, comparing the performance across different similarity kernels (Cho & Saul, 2009) such as: RBF, Cosine, Linear, and Polynomial. The results demonstrate that, across all three datasets, the accuracies are relatively similar, with the RBF kernel exhibiting a slight advantage, achieving 88.7% on PACS, 80.3% on VLCS, and 70.2% on OH. These findings suggest that while the choice of kernel exerts only a minor influence, the RBF kernel consistently yields the most notable improvements across the benchmarks. This consistent advantage motivates our choice of RBF as the default kernel in the main experiment.

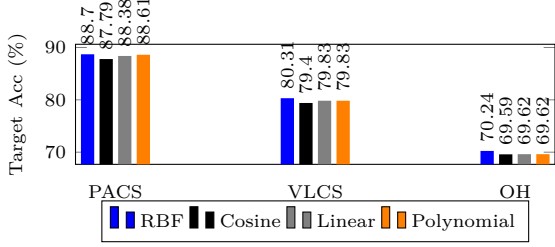 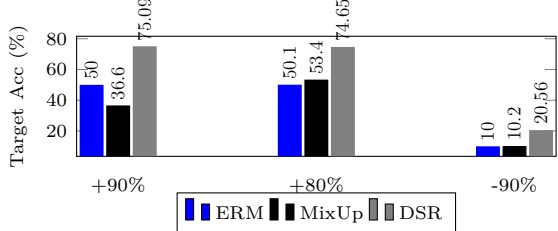

(a) Kernel ablation on PACS, VLCS, and OH (target accuracy).

(b) ColoredMNIST: DG performance of DSR vs baselines.

Figure 8: (a) Kernel ablation across datasets. (b) ColoredMNIST results.

### 5.4.3   Experiments on ColoredMNIST

In this experiment, we utilize ColoredMNIST (Arjovsky et al., 2019), a colored adaptation of the MNIST dataset, frequently employed as a standard benchmark for evaluating DG in the presence of controlled spurious correlations. It includes three domains characterized by different label-color correlations: +90%, +80%, and –90%. As depicted in Figure 8b, DSR, exhibits notable performance enhancements over traditional ERM and MixUp techniques across all domain configurations. Remarkably, DSR surpasses baseline methods by a double margin even in the highly challenging -90% setting, where the correlation is considered to be adversarial. These results emphasize the statistical robustness of DSR in mitigating spurious features and improving generalization across diverse and non-i.i.d. domains.

## 6 Conclusion and Future Works

In this work, we emphasize the underexplored role of diversity in improving DG performance. By utilizing entropy-based uncertainty to guide DPP sampling, we introduce a principled diversity-aware regularization strategy that augments the standard ERM objective. Our method promotes the learning of domain-invariant features without relying on explicit data augmentation or target domain information. Empirically, we demonstrate that our approach not only consistently outperforms both ERM and augmentation-based baselines, but also secures new SOTA results across most of the DG benchmark datasets. These results underscore the effectiveness and simplicity of diversity-driven sample selection in mitigating distribution shifts and improving generalization to unseen domains. We acknowledge, however, that our current experiments are limited by the number of source domains, and increasing this number correspondingly raises the computational cost of DPP based sampling due to its dependence on kernel construction and determinant-based selection. Addressing such a scalability challenge whether through approximate or low-rank DPP formulations, mini-batch level sampling, or more efficient diversity surrogates represents a crucial direction for future research. Overall, these results highlight the effectiveness of diversity-driven sample selection while clearly outlining its current computational limitations and future scalability potential.

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

# A  Appendix

## A.1  Training DSR: Algorithm

The high-level implementation concept of DSR is illustrated in Appendix 1. We have implemented DSR as an auxiliary training objective that functions on intermediate feature representations within a multi-source DG framework. During each training iteration, mini-batches are sampled from all source domains and aggregated. Subsequently, features extracted by the encoder are utilized to compute predictive probabilities and their corresponding entropy. This entropy acts as a measure of model uncertainty and is employed to modulate the feature representations, highlighting samples that are more ambiguous or likely to reside near domain boundaries. A DPP kernel is then constructed over the modulated features, and a log-determinant regularizer is applied to encourage diversity by penalizing redundant representations in the feature space. The resulting diversity regularization term is optimized in conjunction with the standard classification loss in an end-to-end manner, without necessitating explicit domain alignment or additional supervision. This approach implicitly fosters robust and diverse representations that generalize across unseen domains.

---

**Algorithm 1** Training DSR method

---

**Require:** $\{\mathcal{D}_i\}_{i=1}^N$, model $f_\theta$, parameters $\theta$, learning rate $\eta$, regularization weight $\alpha$, epochs $T$, iterations $K$

1: Initialize $\theta$
2: **for** $t = \{1, \ldots, T\}$ **do**
3:   **for** $k = \{1, \ldots, K\}$ **do**
4:     **for** each $\mathcal{D}_i, i \in \{1, \ldots, N\}$ **do**
5:       Sample a mini-batch $\mathcal{B}_i = \{(\mathbf{x}_{ij}, y_{ij})\}_{j=1}^{\mathbf{B}}$ from $\mathcal{D}_i$, $b \in \{1, \ldots, \mathbf{B}\}$
6:     **end for**
7:   **end for**
8:   Concatenate all mini-batches: $\mathcal{B} = \bigcup \mathcal{B}_i$
9:   Generate features $\mathbf{z} = h(\mathcal{B})$, Predict logits $\hat{y} = g(\mathbf{z})$, and probabilities $p = \text{softmax}(\hat{y})$
10:  Compute entropy $\mathbf{u} = -\sum_c p^{(c)} \log p^{(c)}$
11:  Perform Feature Modulation: $\tilde{\mathbf{z}} = \mathbf{z} \cdot \mathbf{u}$
12:  Compute pairwise kernel matrix $\mathbf{L} = \exp(-\gamma \|\tilde{\mathbf{z}}_i - \tilde{\mathbf{z}}_j\|_2^2)$, where $\gamma = (\text{median}\|\tilde{\mathbf{z}}_i - \tilde{\mathbf{z}}_j\|_2^2 + \epsilon)^{-1}$
13:  Compute regularization term: $\mathcal{R}_{\text{div}} = -\log \det(\mathbf{L} + \epsilon I)$
14:  Compute loss: $\mathcal{L}_{\text{total}} = \alpha \cdot \frac{1}{N} \sum_{i=1}^N \mathbb{E}_{(\mathbf{x},y) \sim \mathcal{D}_i}[\ell(f(\mathbf{x}), y)] + (1 - \alpha) \cdot \mathcal{R}_{\text{div}}$
15:  Update parameters: $\theta \leftarrow \theta - \eta \nabla_\theta \mathcal{L}_{total}$
16: **end for**

---

## A.2  How Informative is Domain Sample Size and its Diversity in DSR?

As summarized in Table 1, significant imbalance in sample sizes is common across DG benchmark datasets. Under such conditions, conventional minibatch sampling from a pooled dataset tends to disproportionately favor domains with larger sample sizes. Since random sampling does not account for feature redundancy, minibatches are often dominated by statistically frequent but highly correlated samples. Such a repeated exposure to majority-style features can reinforce spurious correlations. For instance, if images from a "Photo"

domain consistently contain a specific background that is absent in "Sketch" domain, then standard sampling may encourage the model to rely on this background as a shortcut for prediction. Because such features appear frequently in majority-dominated minibatches, the model overfits to domain-specific cues rather than learning domain-invariant representations, resulting in degraded performance on unseen target domains.

In contrast, DSR operates on the pooled sample set but explicitly promotes informative and diverse sample selection through entropy-guided DPP sampling. Instead of ensuring balanced sampling across domains, DSR selects samples that maximize joint diversity in the learned feature space, conditioned on predictive uncertainty. This mechanism implicitly mitigates domain imbalance: samples from underrepresented or more challenging domains tend to exhibit higher uncertainty or contribute novel feature variations, making them more likely to be selected by $\mathcal{R}_{\mathrm{div}}$. Importantly, domain labels themselves are not required by our method. From the perspective of DSR, the objective is not to sample from specific domains, but to select samples that collectively provide maximal information for learning domain-invariant representations. The key factor is not the domain of origin, but whether the samples provide complementary, non-redundant information. In this sense, domain diversity emerges as a byproduct of diversity in the learned representation space, rather than as an enforced structural prior.

### A.3 Cost of DPP Kernel Computation

We conducted a further analysis of the computational overhead introduced by our DPP-based diversity regularization. Specifically, we assessed the additional time needed to compute the DPP kernel and its log-determinant during each training iteration. The observed overheads are reported as in Table 4. It is important to note that all datasets, except DomainNet, consist of four domains, whereas DomainNet contains six domains. Consequently, the computational cost scales moderately with the number of domains involved, as the DPP kernel computation depends on the domain-wise feature dimensionality and pairwise similarity evaluations. Despite this, the additional cost remains relatively minor indicating that the proposed regularization introduces only a small and manageable overhead while enhancing model diversity and generalization.

Table 4: Additional computational cost introduced by DPP kernel and log-determinant computation.

| Dataset | # Domains | # Classes | Time (ms) |
|---|---|---|---|
| PACS | 4 | 7 | 96.1 |
| VLCS | 4 | 5 | 91.6 |
| OfficeHome | 4 | 65 | 91.3 |
| TerraIncognita | 4 | 10 | 92.4 |
| DomainNet | 6 | 345 | 171.3 |

### A.4 Effect of Increasing Batch Size

In our experiment, we explored a variety of batch sizes, specifically 4, 8, 16, 32, 64, and 128. As illustrated in Figure. 9, there is a noticeable trend where increasing the batch size correlates with enhanced model accuracy. This improvement can be attributed to the fact that, with more samples per batch during each epoch, DPP is more likely to sample a wider range of diverse representations from the feature pool.

### A.5 In-Domain Generalization

In Figure. 10, we present a comparison of In-Domain (ID) accuracy with one of the most effective baselines, SWAD (Cha et al., 2021), from our study. To ensure a fair comparison, we used the original implementation of SWAD to replicate their results. The results for SWAD + DSR were obtained by training their method with DSR's objective. It is clear that, with this synergy DSR is effective in maintaining the average ID accuracy across all DG benchmarks. This shows that DSR not only promotes feature diversity but also preserves the underlying representation of domain structure. Such a balanced regularization ensures that

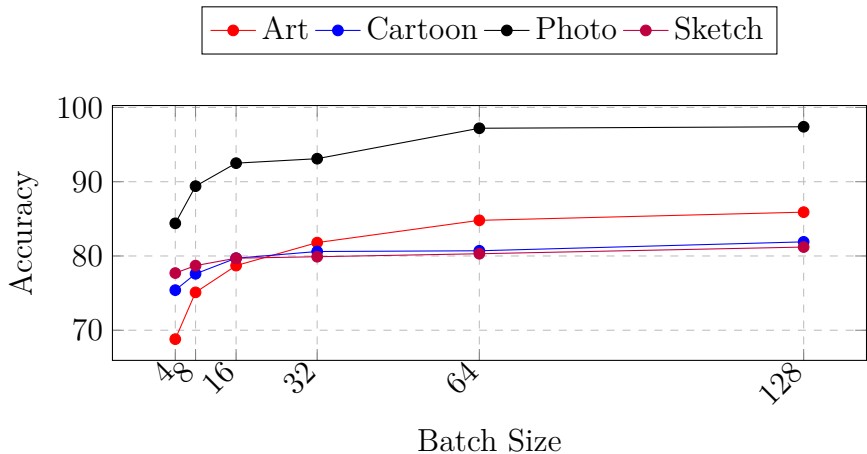

Figure 9: Batch size vs Accuracy comparison of PACS dataset on Resnet-50 architecture.

feature learning remains robust and generalizable, allowing the model to retain competitive ID and OOD performance tradeoffs.

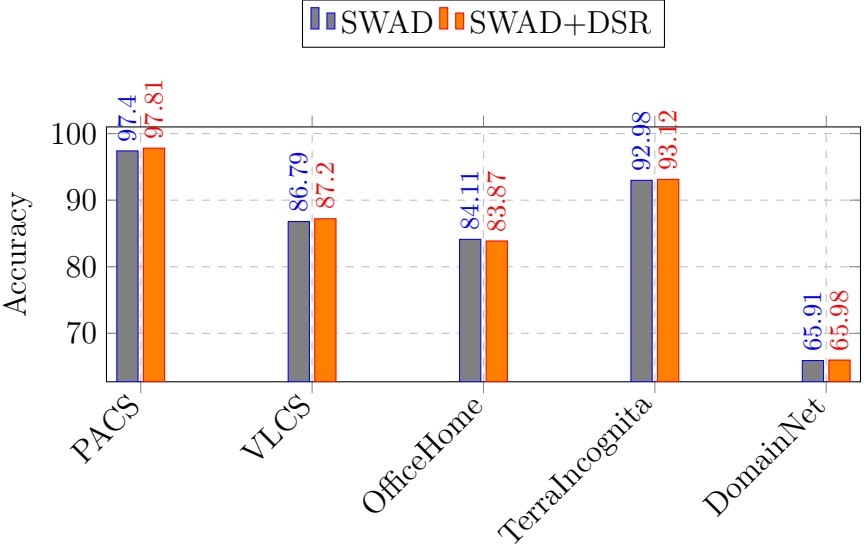

Figure 10: Comparison of In-domain performance of SWAD+DSR and SWAD.

## A.6  Comparison with Diversity Sampling Method

In this section, we compare DSR with DOMI, a closely related approach that utilizes a two-level diversity sampling strategy for domain generalization across multiple domains. However, the original DOMI implementation offers limited information on reproducibility, and its scalability to standard DG benchmarks remains unexplored. Their experiments were confined to the MNIST dataset, so we replicated the setup on the RotatedMNIST benchmark, where digits are rotated by 0,15,30,45,60,75 degrees to simulate multi-domain conditions. Employing a one-domain-out validation strategy, we trained on five rotation domains and evaluated on the remaining one. As shown in Table. 5, DSR consistently achieves higher accuracy across all test domains, with an average accuracy of 97.41%, significantly outperforming both $DOMI_{MMD}$ (87.70%) and $DOMI_{CORAL}$(89.60%). This illustrates that DSR's DPP-based diversity sampling effectively captures a broader and more representative range of feature variations, allowing it to surpass the existing diversity sampling techniques on this benchmark.

Table 5: Comparison of DSR with DOMI baselines on the RotatedMNIST dataset.

| Train Domains (degrees) | Test Domain (degree) | Accuracy (%) |
|---|---|---|
| 0, 15, 30, 45, 60 | 75 | 98.47 |
| 0, 15, 30, 45, 75 | 60 | 95.09 |
| 0, 15, 30, 60, 75 | 45 | 98.71 |
| 0, 15, 45, 60, 75 | 30 | 98.46 |
| 0, 30, 45, 60, 75 | 15 | 98.44 |
| 15, 30, 45, 60, 75 | 0 | 95.30 |
| **Average (DSR)** | – | **97.41** |
| **DOMI$_{\text{MMD}}$** | – | 87.70 |
| **DOMI$_{\text{CORAL}}$** | – | 89.60 |

### A.7 Performance Comparison Across Different Network Architectures

We assess the performance of DSR across three commonly used lightweight neural network architectures: VGG16, ResNet-18, and ResNet-50. The hyperparameters $\gamma=0.05$ and $\alpha=0.5$ remain constant throughout all experiments. As shown in Figure. 11, ResNet-50 consistently achieves the highest accuracy across all datasets, which corresponds to its enhanced representational capacity due to its deeper layers. The figure also underscores that different domains within each dataset exhibit significant variations in accuracy, reflecting the inherent complexity and diversity of visual characteristics across domains. For instance, in VLCS, Caltech yields the highest performance while LabelMe records the lowest, whereas in PACS, Picture achieves the best results and Cartoon the weakest. Despite these differences, the relative performance trends remain consistent across architectures, confirming that the DPP-based diversity regularization in DSR enhances robustness and generalization without requiring additional data generation or augmentation.

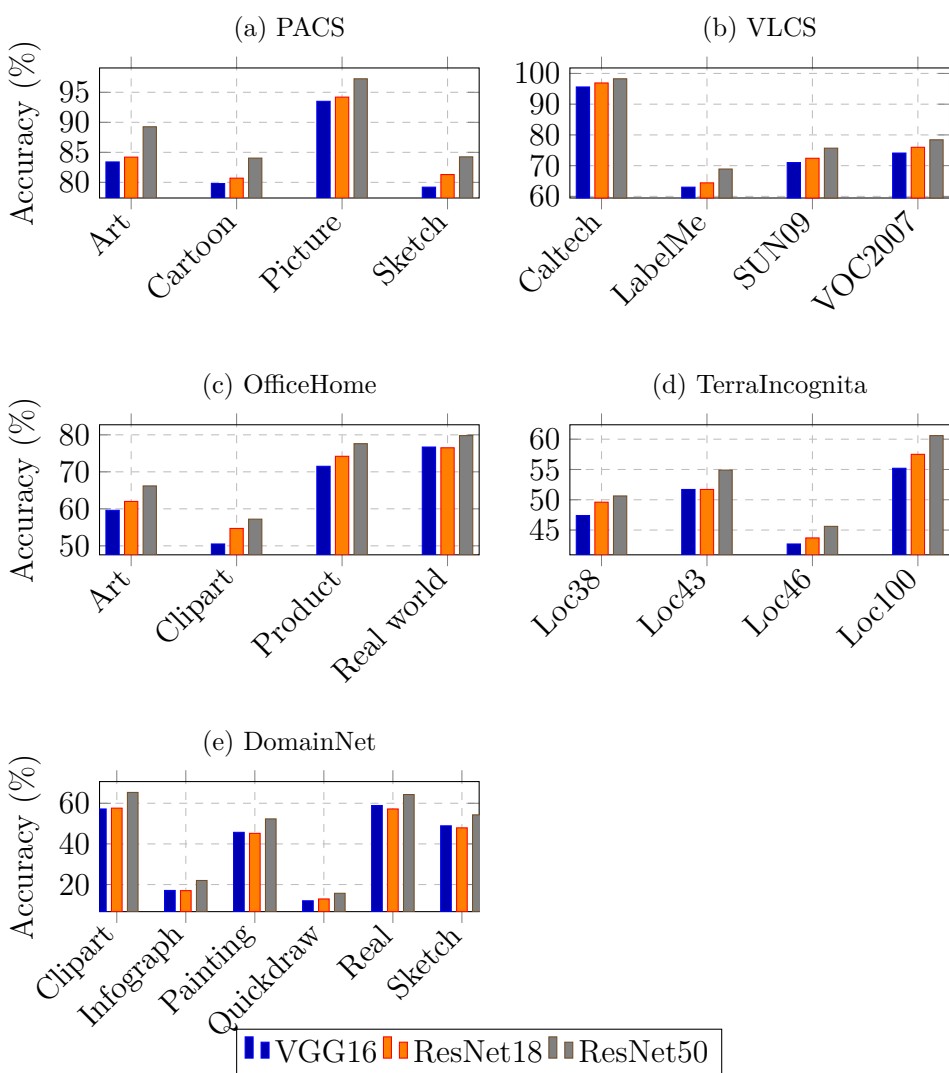

Figure 11: Accuracy comparison across five DG benchmarks using different neural network backbones.

