# OpenReview forum: "Diversity Sampling Regularization for Multi-Domain Generalization"
_TMLR — Accepted by TMLR_

### Review · Reviewer_Qhob · 2025-11-22

**Summary Of Contributions:**

This paper introduces Diversity Sampling Regularization (DSR), a method to improve domain generalization (DG) by promoting diversity in feature representations. DSR leverages predictive entropy to modulate features and applies Determinantal Point Processes (DPP) to sample diverse subsets during training. The approach is model-agnostic and does not rely on data augmentation. Extensive experiments on standard DG benchmarks show that DSR consistently outperforms ERM and several strong baselines, achieving competitive or state-of-the-art results.

**Audience:**

Yes

**Audience Explanation:**

The paper tackles the critical problem of domain generalization, which is of central importance to the machine learning community. Its proposed method is both practical (as a simple, model-agnostic regularizer) and theoretically interesting (leveraging DPPs for diversity). The compelling results across standard benchmarks will attract a broad audience, including researchers focused on robustness, theory, and real-world applications.

**Broader Impact Concerns:**

This work has predominantly positive broader impacts by contributing to more robust and reliable ML systems that can perform well under domain shift, a key challenge for real-world deployment. Potential negatives, such as dual-use or the method's inability to fully address social biases, are common to the field and not uniquely raised by this paper. The authors' approach is resource-conscious compared to augmentation-heavy alternatives, and they transparently report computational costs. The net impact is positive.

**Claims And Evidence:**

Yes

**Claims Explanation:**

Accuracy: Results are reported as mean ± standard error over multiple runs on five standard DG benchmarks, against a wide range of strong baselines and SOTA methods.

Convincing Evidence: The consistent performance gain across all datasets is compelling. Critical ablation studies prove the value of key components (entropy modulation, adaptive γ). The demonstration that DSR boosts the performance of other methods (model-agnosticity) strongly supports its utility as a general regulariz

**Requested Changes:**

How does DSR scale with the number of domains or classes? Is there a risk of performance saturation as diversity increases?

Could DSR be applied to non-visual domains (e.g., NLP or tabular data)?

Have you considered combining DSR with recent sharpness-aware or flatness-based methods (e.g., SWAD) to see if they are complementary?

The entropy modulation seems crucial. Did you experiment with other uncertainty measures (e.g., confidence scores, MC dropout)?

---

> ### Author Response · Authors · 2025-11-26
> **Response to Reviewer Qhob Comments.**
>
> First and foremost, we would like to thank the reviewers in taking part in this review process. We appreciate their valuable time and contribution in helping to make the manuscript better. In the following, we respond to the reviewer's query.
>
> ### 1. How does DSR scale with the number of domains or classes? Is there a risk of performance saturation as diversity increases?
>
> In our research, we have conducted experiments on DSR with the popular DG benchmarks across the literature, especially on the image domain. These datasets have different number of domains and classes as shown below:
>
> | Dataset        | Domains | Classes |
> |----------------|---------|---------|
> | PACS           | 4       | 7       |
> | VLCS           | 4       | 5       |
> | OfficeHome     | 4       | 65      |
> | TerraIncognita | 4       | 10      |
> | DomainNet      | 6       | 345     |
> Among these, OfficeHome and DomainNet are the most challenging due to their large label spaces, while DomainNet is additionally difficult because it contains six heterogeneous domains. This naturally increases the difficulty of representation learning for all DG algorithms, including that of DSR. Empirically, we observe that this scaling behavior is consistent across existing methods, and DSR follows the same trend. However more importantly DSR is designed as an augmentive objective that can be plugged into any backbone method. And across all small and large datasets, DSR consistently provides positive performance gains over the corresponding vanilla baselines.
>
> For example: DSR + ERM achieves +3.8% improvement on PACS and a larger +5.95% improvement on the more complex DomainNet. This demonstrates that DSR continues to deliver meaningful gains even as the diversity of domains and classes increases, without performance saturation.
>
> Regarding computational impact: as reported in Table 3 (Appendix), the relative overhead grows with dataset complexity. For instance, training on DomainNet is roughly twice as slow as on PACS due to more domains to sample from and its considerably larger class space. Nevertheless, the performance improvement does not deteriorate, and we did not observe cases where increased diversity harms DSR’s effectiveness.
>
> ### 2. Could DSR be applied to non-visual domains (e.g., NLP or tabular data)?
>
> Yes. DSR is fundamentally modality-agnostic because it operates on diversity-aware sampling in the feature space, independent of visual structure. In principle, the similar sampling mechanism could be applied to NLP (using contextual text embeddings) and tabular data (using learned or hand-crafted representations). However, our current experiments focus on the image domain because standardized and widely used DG benchmarks for NLP and tabular data remain limited. While datasets such as FMoW-Tabular and the recent Tabular DG benchmark exist, they are not yet as mature or broadly adopted as the image-based DG benchmarks, making systematic comparison challenging. Nonetheless, DSR’s formulation is directly extendable to non-visual domains, and we consider applying it to NLP and tabular DG as a promising direction for future work as more robust benchmarks become available.
>
> ### 3. Have you considered combining DSR with recent sharpness-aware or flatness-based methods (e.g., SWAD) to see if they are complementary?
> Yes, we have considered that and we have presented the comparison of DSR + SWAD on ID accuracy in Fig.10 Appendix. We also have the results which were not considered due to space constraint. However, considering the reviewer's suggestion we will update our manuscript with these results and include the comparison. We have revised the manuscript with this update. In Fig. 5(f), we show the performance comparison of SWAD and SWAD+DSR in a bar graph. We added this figure in the existing comparisons of vanilla and DSR augmented methods. For the reference of reviewer, we present the result of SWAD and DSR as follows.
>
> | Dataset        | SWAD Avg | SWAD + DSR Avg |
> |----------------|----------|----------------|
> | PACS           | 88.19    | 88.53          |
> | VLCS           | 78.77    | 79.15          |
> | OfficeHome     | 70.50    | 70.67          |
> | TerraIncognita | 47.50    | 49.00          |
> | DomainNet      | 46.25    | 46.40          |
>
> ### 4. The entropy modulation seems crucial. Did you experiment with other uncertainty measures (e.g., confidence scores, MC dropout)?
> Confidence-based metrics becomes unstable because a single value (e.g., max softmax probability) might reflect only the model’s top prediction, whereas entropy accounts for the entire class distribution and provides a smoother, more reliable uncertainty measure. MC Dropout, while feasible, requires multiple stochastic forward passes with dropout active during both training and inference, which is computationally impractical for large-scale DG. Considering these factors, we chose entropy as an uncertainty measure offering the optimal balance of stability and efficiency for DSR.

---

### Review · Reviewer_ws11 · 2025-11-27

**Summary Of Contributions:**

This paper presents DSR, a method for Domain Generalization designed to improve performance on unseen domains by addressing the limited feature diversity often observed in traditional augmentation techniques. The approach employs entropy values to assess input prediction uncertainty, using these scores to guide a Determinantal Point Process that selects diverse data subsets near the decision boundary. By integrating this sampling process as a regularization term within the ERM framework, the method seeks to learn domain-agnostic features without explicit data synthesis. Empirical evaluations across benchmarks such as PACS, VLCS, and DomainNet show that DSR yields improvements over baselines.

Strengths:
1. The method combines entropy-aware sampling and pairwise feature distances to encourage subset diversity, is lightweight to implement, and achieves leading or competitive results on several benchmarks.
2. DSR is formulated as an architecture-agnostic, plug-and-play regularizer, making it straightforward to integrate into a variety of existing DG frameworks and backbone models.

Weaknesses:
1. Some design choices would benefit from stronger, more principled motivation. In particular, entropy-based feature scaling adopts heuristic priors that may limit interpretability and raise concerns about sensitivity to these assumptions.
2. On benchmarks such as OfficeHome and TerraIncognita, DSR underperforms SWAD by a notable margin. This raises the question of whether certain inductive priors in DSR may be better aligned with specific datasets, and whether a more prior-free or adaptive formulation could improve robustness across a broader set of domains.

**Additional Comments:**

N/A

**Audience:**

Yes

**Audience Explanation:**

The topic of Domain Generalization is a classic and critical problem in machine learning. It addresses the challenge of building models that remain robust when faced with changes in data distribution or domain shifts, which is essential for successful deployment in diverse real-world application scenarios

**Claims And Evidence:**

Yes

**Claims Explanation:**

The evidence includes DSR consistently outperforming ERM and MixUp, demonstrating model agnosticity by boosting the performance of existing methods, and confirming the utility of its key components through targeted ablation studies.

**Requested Changes:**

1. It would be valuable to provide an analytical investigation into why the proposed DSR method underperforms compared to SWAD on the OfficeHome and DomainNet benchmarks. One potentially insightful direction would be to visualize and compare the feature distributions of the learned representations from DSR and SWAD, which may help reveal why DSR’s feature-space geometric constraints could be suboptimal for these datasets.

2. Is the method limited to classification tasks, or can it be extended to more complex settings, such as segmentation and object detection?

---

> ### Author Response · Authors · 2025-12-01
> **Response to Reviewer ws11 Comments.**
>
> First and foremost, we would like to thank the reviewers in taking part in this review process. We appreciate their valuable time and contribution in helping to make the manuscript better. In the following, we respond to the reviewer's query.
>
> ## Requested Changes:
>
> 1. It would be valuable to provide an analytical investigation into why the proposed DSR method underperforms compared to SWAD on the OfficeHome and DomainNet benchmarks. One potentially insightful direction would be to visualize and compare the feature distributions of the learned representations from DSR and SWAD, which may help reveal why DSR’s feature-space geometric constraints could be suboptimal for these datasets.
>
> In the paper, we have shown DSR being a simple ERM framework augmented with DPP diversity sampling objective. In addition to that, the nature of DSR is method agnostic where it can be augmented with learning objective of any methods. This means we can also do SWAD + DSR. For the reference, below table shows the results of vanilla SWAD and its counterpart SWAD+DSR. We can see that, with DSR attached to the learning objective, SWAD is able to enhnace its DG accuracy on all benchmarks.
>
> | Dataset        | SWAD Avg | SWAD + DSR Avg |
> |----------------|----------|----------------|
> | PACS           | 88.19    | 88.53          |
> | VLCS           | 78.77    | 79.15          |
> | OfficeHome     | 70.50    | 70.67          |
> | TerraIncognita | 47.50    | 49.00          |
> | DomainNet      | 46.25    | 46.40          |
>
> We have revised the manuscript with this update. In Fig. 5(f), we show the performance comparison of SWAD and SWAD+DSR in a bar graph. We added this figure in the existing comparisons of vanilla and DSR augmented methods.
>
> 2. Is the method limited to classification tasks, or can it be extended to more complex settings, such as segmentation and object detection?
>
> Thank you for pointing this out. We certainly believe that DSR's extension to more complex settings such as segmentation and object detection would bring more practicality of the proposed method. At this stage, we have limited our scope to image classification tasks only because much of DG benchmarks are readily available and popular in this domain. We certainly plan to examine how DSR performs in other domains. Since DSR specifically targets the learning dynamics of the method by sampling diverse representations, we believe that transferring DSR to other modalities and downstream applications is possible, given that we have labeled datasets. For now, we keep this extension for the future works.

---

### Review · Reviewer_Xe9f · 2025-12-19

**Summary Of Contributions:**

This paper provides a method for domain generalization based on regularization via diversity sampling, which roughly works by prioritizes sampling data that has more uncertain prediction values. The authors provide extensive experimental validation to show the improvements of their method.

Strengths:
- The experiments show that the method beats state of the art methods

Weaknesses:
- The presentation is a little hard to follow, e.g. some details are missing.

**Additional Comments:**

Typos:
- Page 2: In this paper, we propose... "could" should be replaced with "to"?
- Section 3.8: 4 -> (4) as eqref
- 1e-2 -> 0.01?

Minor:
- Section 3.1: $\mathcal{F}$ should be defined to be some function class
- Section 3.2: 2a -> Figure 2a, similarly 2b -> Figure 2b (and other instances where figures and tables are referenced)

Questions:
- $\mathcal{C}$ is not defined in Eq. 3. Should we think of this as (average/max) TV/Wasserstein/KL bound?
- $\mathcal{R}_{div}$ is also not defined. It should be stated that this can be chosen by the practitioners, and also give some suggestions on how to choose it (e.g. in appendix).
- What is z in Eq (5)? All I can find is based on Figure 3(a)
- $\alpha = 0.5$ seems to put quite a large fraction of the weight on the regularizer. I'm not sure how reasonable this is in practice, as one would expect regularization to be small (like between 0.1 and 0.2). Can the authors discuss good choices for $\alpha$?

**Audience:**

Yes

**Audience Explanation:**

Domain generalization is an important method in machine learning, as it doesn't require gathering additional data on unseen domains for (re)training.

**Claims And Evidence:**

Yes

**Claims Explanation:**

Yes, for the most part. The main theoretical claim is supported by evidence.

One minor issue is that I don't think tsne is a valid way to compare inter-class separation on datasets (Figure 4). Even rerunning tsne on the same data (with a different seed) will lead to vastly different results.

**Requested Changes:**

It would strengthen the paper if an appendix describing how DPP works, to make the work self-contained. Additionally, with just reading the paper in its current state, I do not think I could implement the algorithm. It may be helpful to add an algorithm environment and/or reformat the discussion to make it easier to understand a high-level implementation of the method.

---

> ### Author Response · Authors · 2025-12-23
> **Response for reviews for Reviewer Xe9f**
>
> First and foremost, we would like to thank the reviewer in taking part in this review process. We appreciate their valuable time and contribution in helping to make the manuscript better. In the following, we respond to the reviewer's query.
>
> ## Correction of typos and minor erratas.
>
> We have corrected all the typos and minor mistakes as specified by the reviewer. We reflect this changes in the updated version of the manuscript.
>
> ## Questions.
> - $\mathcal{C}$ is not defined in Eq. 3. Should we think of this as (average/max) TV/Wasserstein/KL bound?
>
> $\mathcal{C}$ is a domain discrepancy term and it can be regarded as any of the integral probability metrics such as Wassertian or KL bound.
>
> We have updated the manuscript with the following text:
>
> > Here, $\mathcal{C}$ denotes a generic domain discrepancy term that measures the distributional shift between the collection of source domains and the target domain, and can be instantiated by any divergence or distance for which a risk transfer bound of the form in (\ref{eq:3}) holds (e.g., integral probability metrics (IPM) or $f$ divergence based measures), depending on the underlying assumptions on the data distributions, loss function, and hypothesis class. Note that, throughout this work, we treat $\mathcal{C}$ abstractly and do not assume a
> specific form, as our method aims to reduce domain discrepancy implicitly via representation diversity rather than optimizing an explicit divergence.
>
>
> - $\mathcal{R}_{\text{div}}$ is also not defined. It should be stated that this can be chosen by the practitioners, and also give some suggestions on how to choose it (e.g. in appendix).
>
> In our method, $\mathcal{R}\_{\text{div}}$ is the diversity regularizer and it is the main term in the manuscript. In section, 3.6, we have explicity defined and discussed how $\mathcal{R}_{\text{div}}$ can be obtained. For clarity, we have updated the title of section 3.6 to following.
>
> > Reducing Domain Discrepancy through Diversity Regularizer $\mathcal{R}_{\text{div}}$
>
> - What is z in Eq (5)? All I can find is based on Figure 3(a)
>
> z is the feature vector. We have now defined it where it first appears in the manuscript. The update is as follows:
>
> > As shown in the block diagram in Figure. 3a, DSR leverages predictive uncertainty followed by DPP sampling to induce diversity constraint among feature vectors $\mathbf{z}$, where $\mathbf{z} = h(\mathbf{x}) \in \mathbb{R}^d$.
>
> - $\alpha = 0.5$ seems to put quite a large fraction of the weight on the regularizer. I'm not sure how reasonable this is in practice, as one would expect regularization to be small (like between 0.1 and 0.2). Can the authors discuss good choices for $\alpha$?
>
> We have performed ablation study on the value of $\alpha$ as shown in Figure. 7b. Here, we can observe that the target accuracy on all DG benchmarks are optimal at nearby the value of alpha = 0.5. As the value increases, there is slight performance drop. Across small regularization strength like 0.1, and 0.2, the performance is still better than the larger values of alpha > 0.5.
>
> ## Requested change
>
> On the request of the reviewer, we have updated the new version of the manuscript with the high level implementation idea presented as an algorithm. We have placed the algorithm in the appendix of the manuscript.

---

### Decision · Action_Editor_7Fho · 2026-01-20

**Recommendation:** Accept with minor revision

**Additional Comments:**

### Requested Changes:

1. Based on Eq. (4) and Algorithm 1 in Sec. A.1, each domain is treated as equally important by default. The distinction between DPP + ERM and treating all domains as a single pool is not obvious, leaving the role and impact of domain labels unclear. A discussion on how $\mathcal{R}_{div}$ is affected by domain diversity and varying sample sizes across domains would be valuable.

2. In Fig. 3(a), there are two "Low Entropy"s; one of them should be "High Entropy."

3. Since the current experiments involve at most six domains, the scalability issue to a much larger number of domains remains. Discussing potential solutions to this limitation would be beneficial.

**Audience:**

Yes

**Audience Explanation:**

Domain generalization enables models to remain robust on unseen domains without additional data, which is critical for real-world applications. This topic is of significant interest to the TMLR community.

**Claims And Evidence:**

Yes

**Claims Explanation:**

The claims made in the submission are generally well-supported by evidence. The proposed method provides an architecture-agnostic, plug-and-play regularizer, making it straightforward to integrate into a variety of existing domain generalization (DG) frameworks and backbone models. Empirical evaluations on benchmarks such as PACS, VLCS, and DomainNet demonstrate that DSR consistently outperforms baseline methods.

In response to the initial reviews, the authors have updated the following to address related concerns:
1. Empirical study of scalability across domains and classes.
2. Discussion for extension to more complex settings.
3. Integration with SWAD, including new combined results.
4. Justification for the choice of entropy over other uncertainty measures.
5. Clarity and reproducibility, through updated notations, a detailed algorithm, and self-contained descriptions.

Although concerns remain regarding the extension of these results to modern vision or multimodal models, addressing them is not essential for the scope of the current study.